# ON THE DECOMPOSITION OF DIFFERENTIABLE GAMES

## ABSTRACT

To understand the complexity of the dynamic of learning in Differentiable games, we decompose the game into components where the dynamic is well understood. One of the possible tools is Helmholtz's theorem, which can decompose a vector field into a potential and a harmonic component. This has been shown to be effective in finite and normal-form games. However, applying Helmholtz's theorem by connecting it with the Hodge theorem on $\mathbb{R}^n$ (which is the strategy space of the Differentiable game) is non-trivial due to the non-compactness of $\mathbb{R}^n$. Bridging the dynamic-strategic disconnect through Hodge/Helmholtz's theorem in Differentiable games is then left as an open problem in Letcher et al. (2019). In this work, we provide two decompositions of Differentiable games to answer this question: the first as an exact Potential part, a near Solenoidal part, and a non-strategic part; the second as a near Potential part, an exact Solenoidal part, and a non-strategic part. We show that Potential games coincide with potential games proposed by Monderer & Shapley (1996), where the gradient descent dynamic can successfully find the Nash equilibrium. For the Solenoidal game, we show that the individual gradient field is divergence-free, in which case the gradient descent dynamic may either be divergent or recurrent.

## 1 INTRODUCTION

One of the most fundamental questions in game-theoretic learning is whether an uncoupled learning dynamic can ultimately achieve a stable equilibrium through repeated interactions among players. Specifically, in which games and under what conditions can players reach a stable state using learning algorithms such as gradient descent? This issue has gained significant attention, particularly as many advancements in machine learning have relied on gradient descent to optimize the parameters of neural networks, with objective functions that model non-cooperative games. Popular examples include adversarially generative networks Goodfellow et al. (2020), federated learning Kairouz et al. (2021); Donahue & Kleinberg (2021), multi-agent reinforcement learning Littman (1994); Lowe et al. (2017), and any machine learning algorithm trained in an adversarial way.

A notable result by Hart & Mas-Colell (2003; 2006) presents a negative finding, demonstrating that no uncoupled learning dynamics can converge to a Nash equilibrium in all games from any initial condition. This raises the critical question of which games a learning process can successfully converge to a Nash equilibrium and which games it cannot.

In the case where the game is finite, or if the game is a normal-form game, this question can be partially answered by decomposing the game through Helmholtz's theorem. When the number of strategies for each player is finite, or when the strategies considered are on a probability simplex, Candogan et al. (2010); Legacci et al. (2024) showed that a game can be decomposed into a potential game and a harmonic game. This decomposition categorizes finite games along a spectrum, ranging from players with fully aligned interests (represented by games containing only the potential component) to players with entirely conflicting interests (represented by games containing only the harmonic component). In potential games, where there exists a potential function to quantify how individual strategy changes affect collective utility, players can thus descent along the direction of the gradient of their utility functions, which is equivalent to collectively minimizing the potential function, to reach the Nash equilibrium. In normal form games, the harmonic game is shown to be

incompressible, hence implying that common learning dynamics, such as the exponential weight, can lead to Poincaré recurrence Legacci et al. (2024).

In Differentiable games, where the strategies are assumed to be in $\mathbb{R}^n$, applying Helmholtz's theorem becomes non-straightforward. Different from the normal form games, where the utilities are multilinear and the strategies are naturally in a compact set, the utility of Differentiable games can be much more complicated. Specifically, Helmholtz's theorem operates in $\mathbb{R}^3$, where the curl of the gradient field is still a vector field, which means naively applying Helmholtz's theorem only gives a decomposition in $\mathbb{R}^3$. When $n \geq 4$, the curl of the gradient field is no longer a vector field, and one then needs to leverage the Hodge Theorem to perform the decomposition on a manifold. Connecting and applying Hodge/Helmholtz decomposition on the manifolds is yet to be investigated. A direct sum decomposition, like those in finite games and normal-form games, remains open in Differentiable games Letcher et al. (2019).

## 1.1 RELATED WORKS

In finite games, where the number of strategies is finite for each player, Candogan et al. (2011) introduced a method to decompose a given game into a potential and harmonic component. This decomposition maps finite games into a spectrum of players having fully aligned interests (games with only the potential component), to players having completely conflicting interests (games with only the harmonic component). Follow-up works then develop different variants of decompositions for the finite games Cheng et al. (2016); Wang et al. (2017); Li et al. (2019); Abdou et al. (2022).

This decomposition framework is then extended to normal form games Legacci et al. (2024), where classic no-regret algorithms such as exponential weights are known to be possibly chaotic Palaiopanos et al. (2017); Mertikopoulos et al. (2018); Vlatakis-Gkaragkounis et al. (2019). Based on the decomposition of normal form games, Legacci et al. (2024) provided a principled way to identify cycling behaviors of exponential weights.

In Differentiable games, Letcher et al. (2019) identified two classes of games based on the symmetric and skew-symmetric parts of the game's Jacobian matrix. The games with the symmetric Jacobian matrix are identified to be potential games, while games with skew-symmetric Jacobian matrix are named Hamiltonian games, which are closely related to harmonic games. However, this is different from the direct sum decomposition results in finite and normal-form games. Specifically, given a game with individual gradient field $g$, it is impossible in general to find a potential game $g_p$ and a Hamiltonian game $g_h$ such that $g = g_p + g_h$. Classic gradient descent methods are known to be convergent for potential games, but they can be non-convergent for Hamiltonian games. They thus introduced symplectic gradient adjustment to find stable fixed points in Differentiable games. They also remarked that connecting the differential-geometric Hodge/Helmholtz decomposition in Differentiable games is left as an open problem.

## 1.2 DIFFERENTIABLE GAMES

We consider a Differentiable game with $M$ players. Each player $i$ has utility $\{u_i : \mathbb{R}^n \to \mathbb{R}\}_{i=1}^{M}$ and can play a strategy $\omega_i \in \mathbb{R}^{n_i}$ to maximize its utility. We denote the joint strategy as $\omega = (\omega_1, \ldots, \omega_M) \in \mathbb{R}^n$ where $\sum_{i=1}^{M} n_i = n$. We also let $\omega_{-i}$ be the joint strategy of all players except for player $i$. To denote the individual components of $\omega$, we write $\omega = (\omega_1, \ldots, \omega_M) = (x_1, \ldots, x_n)$.

## 1.3 OUR CONTRIBUTIONS

We identified two different inner products on the space of Differentiable games, which allows us to apply the Helmholtz decomposition on the vector field space of the utility gradient. Similar to the decomposition of the finite games, we decompose the Differentiable games into three parts, which enjoy different dynamic properties. However, different from the case of finite games, the Differentiable games cannot be decomposed straightforwardly into a potential part and a harmonic part. This is due to the fact that the harmonic component is isomorphic to the de Rham cohomology of the manifold, which is zero when the differential $k$-form is with $k = 1$ and the manifold is $\mathbb{R}^n$. Instead, we identify a Solenoidal part of the Differentiable games, which are similar to a harmonic game in many ways, such as the challenges it imposes on dynamical systems induced by gradient descent.

Our two decompositions provide different interpretations of the space of Differentiable games. In the first decomposition, the game is decomposed into an exact Potential part, a near Solenoidal part, and a non-strategic part. We show that the exact Potential part of the game is an exact potential game, and the Solenoidal part poses similar challenges to learning algorithms. Specifically, the standard gradient descent dynamic can exhibit non-convergent behaviors on the Solenoidal game. To relate the Solenoidal games and the Hamiltonian games, we show that a Hamiltonian game has to be a Solenoidal game, but not vice versa. In the second decomposition, the game is decomposed into a near Potential part, an exact Solenoidal part, and a non-strategic part. We show that the Solenoidal part of the game is flat on any local Nash equilibrium, which imposes significant challenges to first and second-order local Nash equilibrium finding algorithms.

## 2 MAIN RESULTS

### 2.1 OVERVIEW OF RESULTS

Before we present the technical results, we first provide an overview of our main result. In this work, we provide two decomposition methods for Differentiable games. In the first decomposition, the game can be decomposed into an exact Potential game and a near Solenoidal game. In the second decomposition, the game can be decomposed into a near Potential game and an exact Solenoidal game. We show that the Potential game coincides with the canonical notion of potential game, where there exists a potential function to quantify how individual strategy changes affect collective utility.

In physics, the Helmholtz decomposition theorem states that in $\mathbb{R}^3$, a vector field can be decomposed into an irrotational (curl-free) component and a solenoidal (divergence-free) component, known as the scalar potential and vector potential, respectively. Our decomposition directly applies to the simultaneous gradient field of the game, extending to the space $\mathbb{R}^n$ (not limited to $\mathbb{R}^3$). In this context, the irrotational field corresponds to the electric field, while the solenoidal field corresponds to the magnetic field. We remark that in finite games, since every game's pairwise comparison function lies in the kernel of the curl operator (as shown in Candogan et al. (2011)), the solenoidal component does not exist.

The two decompositions can be summarized by the following informal claim.

**Theorem 2.1** (Informal statement combining Theorem 2.3 and Theorem E.1)**.** *The simultaneous gradient $Du = (\nabla_{\omega_1} u_1, \ldots, \nabla_{\omega_M} u_M)$ can be decomposed as $Du = X_{\mathcal{P}} + X_{\tilde{\mathcal{S}}} = X_{\tilde{\mathcal{P}}} + X_{\mathcal{S}}$, where $\mathcal{P}$ is the class of exact Potential games, $\mathcal{S}$ is the class of exact Solenoidal games, $\tilde{\mathcal{P}}$ is the class of near Potential games and $\tilde{\mathcal{S}}$ is the class of near Solenoidal games.*

**Remark 2.1** (Naming of the components)**.** *The name Solenoidal game is given after the fact that a solenoid generates a magnetic field, which is a divergence-free field. This agrees with the characterization of Solenoidal games which induces divergence-free individual gradient field. We believe that a more proper name is "vector potential games" by the above physics inspiration. However, as the term "potential game" is well-established in game theory, we retain "potential" for the scalar potential while renaming the vector potential as the Solenoidal games.*

For the Solenoidal game, we show that the vector field of the gradient is divergence-free. As an implication of that, algorithms based on the gradient descent dynamic may be hard to converge. Specifically, we show that without a specific initialization requirement, the gradient descent dynamic can drift to infinity, in which case the orbits are not bounded, or the induced trajectory may be chaotic. In this case, while it is possible to pick a good initialization point, there is no principle on how to pick such a point in general. This thus provides insights into the difficulty of learning Differentiable games, as we describe in the below claim.

**Theorem 2.2** (Informal statement combining Theorem 2.6 and Theorem E.2)**.** *When a game is in $\tilde{\mathcal{S}}$ and the utility function has compact support, the gradient descent dynamic is Poincaré recurrent. When a game is in $\mathcal{S}$, and the orbits are bounded, the gradient descent dynamic is Poincaré recurrent. When the dynamic is Poincaré recurrent, for almost every initialization, the induced trajectory $x(t)$ returns arbitrarily close to the initial point infinitely often.*

Therefore, our decomposition is not merely a direct sum decomposition in the inner product sense; rather, the two types of games obtained through this decomposition exhibit fundamentally different behaviors from a dynamical systems perspective.

## 2.2 Notations and background

Let $L_{\text{loc}}^1(\mathbb{R}^n)$ denote the set of locally integrable functions on $\mathbb{R}^n$. Let $C^\infty(\mathbb{R}^n)$ denote the space of infinitely differential functions $f : \mathbb{R}^n \to \mathbb{R}$. We let the subscript $c$ denote all the compactly supported functions, i.e. $C_c^\infty(\mathbb{R}^n)$ denotes the space of compactly supported, infinitely differential functions on $\mathbb{R}^n$.

In the context of infinite-dimensional normed linear spaces, Hilbert spaces possess a direct sum decomposition related to the inner product. Therefore, to ensure that our considered spaces are Hilbert spaces (which are complete inner product spaces), we need to use the concept of weak derivatives.

**Definition 2.1** (Weak derivative, Evans (1998)). *Suppose $u, v \in L_{\text{loc}}^1(\mathbb{R}^n)$ and $\alpha$ is a multiindex. We say that $v$ is the $\alpha^{th}$-weak partial derivative of $u$, written $D^\alpha u = v$, provided $\int_{\mathbb{R}^n} u D^\alpha \phi dx = (-1)^{|\alpha|} \int_{\mathbb{R}^n} v\phi dx$ for all $\phi \in C_c^\infty(\mathbb{R}^n)$.*

When the derivative of $f$ exists, it is equivalent to the weak derivative. The introduction of weak derivatives is solely to guarantee the completeness of our inner product space. To get an intuitive understanding of the weak derivative, we use the following example as an illustration.

**Definition 2.2** (Sobolev Space, Evans (1998)). *The Sobolev space $H^k(\mathbb{R}^n)$ consists of all locally summable functions $f : \mathbb{R}^n \to \mathbb{R}$ such that for each multiindex $\alpha$ with $\alpha \leq k$, $D^\alpha u$ exists in the weak sense and belongs to $L^2(\mathbb{R}^n)$. Moreover, if $f \in H^k(\mathbb{R}^n)$, we define its norm to be $\|f\|_{H^k} = \left(\sum_{\alpha \leq k} \int_{\mathbb{R}^n} |D^\alpha f|^2 dx\right)^{1/2}$, where $D^\alpha f$ is the $\alpha th$-weak partial derivative of $f$.*

## 2.3 Decomposition of Differentiable Games

We follow a similar decomposition roadmap to that of finite games and normal form games Candogan et al. (2011); Legacci et al. (2024). However, different from Differentiable games, the utility functions of normal-form games are multilinear and the strategy space is compact, which makes it significantly easier to decompose. To complete our decomposition, we provide a description of the gradient vector field of the utility function, which is with a newly introduced inner product and norms.

To begin with, we first introduce three spaces, $C_0, C_1, C_2$, and operators $d_1, d_2$. The domains and codomains of $d_1$ and $d_2$ are summarized as, $C_0 \xrightarrow{d_1} C_1 \xrightarrow{d_2} C_2$. We then provide a decomposition of $C_1$, which captures the description of the gradient vector field of the game, based on the Helmholtz decomposition.

We define the spaces $C_0, C_1, C_2$ as follows. We also define the operator that maps between $C_0$ and $C_1$ as $d_1$, and the one between $C_1$ and $C_2$ as $d_2$.

**Definition 2.3.** *The Hilbert space $C_0$ is defined as $C_0 = \overline{C^\infty(\mathbb{R}^n)}^{\|\cdot\|_{H^2}}$, which is the closure of the space $C^\infty(\mathbb{R}^n)$ with norm $\|\cdot\|_{H^2}$. The inner product is defined as $\langle f, g\rangle_0 = \int_{\mathbb{R}^n} f(x)g(x)dx + \sum_{i=1}^n \int_{\mathbb{R}^n} \frac{\partial f}{\partial x_i} \cdot \frac{\partial g}{\partial x_i} dx + \sum_{i=1}^n \sum_{j=1}^n \int_{\mathbb{R}^n} \frac{\partial^2 f}{\partial x_i \partial x_j} \cdot \frac{\partial^2 g}{\partial x_i \partial x_j} dx$. The norm is thus $\|f\|_0 = [\langle f, f\rangle_0]^{1/2}$.*

To ensure a simpler notation, we use $x_i$ to denote the integrals and derivatives with respect to the $i$-th component of the joint strategy. We remark that $\frac{\partial f}{\partial x_i}$ is the weak derivative with respect to $f$ on each component of possible joint strategies $\omega$.

**Definition 2.4.** *The Hilbert space $C_1$ is defined as $C_1 = \left\{X = (f_1, \ldots, f_n) \mid f_i \in H^1(\mathbb{R}^n)\right\}$. The inner product is defined as $\langle X, Y\rangle_1 = \sum_{i=1}^n \int_{\mathbb{R}^n} f_i \cdot g_i dx + \sum_{i=1}^n \sum_{j=1}^n \int_{\mathbb{R}^n} \frac{\partial f_i}{\partial x_j} \cdot \frac{\partial g_i}{\partial x_j} dx$. The norm is thus $\|X\|_1 = [\langle X, X\rangle_1]^{1/2}$. The operator $d_1 : C_0 \to C_1$ is defined as follows. For $f \in C_0$, $d_1 f = \left(\frac{\partial f}{\partial x_1}, \ldots, \frac{\partial f}{\partial x_n}\right)$.*

**Definition 2.5.** *The Hilbert space $C_2$ is defined as $C_2 = \left\{ \mathcal{F} \mid (\mathcal{F})_{ij} = \begin{cases} f_{ij} & i < j \\ 0 & i \geq j \end{cases}, f_{ij} \in L^2(\mathbb{R}^n), \forall i, j \in [n] \right\}$. The inner product is defined as $\langle \mathcal{F}, \mathcal{G} \rangle_2 = \sum_{i<j} \int_{\mathbb{R}^n} f_{ij} \cdot g_{ij} dx$. The norm is thus $\|\mathcal{F}\|_2 = [\langle \mathcal{F}, \mathcal{F} \rangle_2]^{1/2}$. The operator $d_2 : C_1 \to C_2$ is defined as follows. For $X = (f_1, \ldots, f_n)$, $(d_2 X)_{ij} = \begin{cases} -\frac{\partial f_i}{\partial x_j} + \frac{\partial f_j}{\partial x_i} & i < j \\ 0 & i \geq j \end{cases}$.*

We now connect these definitions to the class of Differentiable games. We first consider Differentiable games with a class of utility function $u \in C_0^M$, where $C_0^M = \{u = (u_1, \ldots, u_M) \mid u_i \in C_0 \cap C^\infty(\mathbb{R}^n), \forall i\}$. We then define the operator $D$, such that $D_m u = (0, \ldots, \nabla_{\omega_m} u_m, \ldots)$, $Du = \sum_{m=1}^M D_m u = (\nabla_{\omega_1} u_1, \ldots, \nabla_{\omega_M} u_M)$.

**Theorem 2.3.** *$C_1$ can be decomposed as $C_1 = \ker(d_2) \oplus \ker(d_2)^\perp$. Then $Du = X_{\mathcal{P}} + X_{\tilde{\mathcal{S}}}$, where $X_{\mathcal{P}} \in \ker(d_2)$, $X_{\tilde{\mathcal{S}}} \in \ker(d_2)^\perp$.*

To obtain this theorem, we first show that the gradient vector field is in $C_1$. Then we show that $d_2$ is a bounded linear operator, which is guaranteed by leveraging the inner products defined on $C_0$, $C_1$. The proofs are deferred to the appendix.

Now we can define the following classes of subgames, which can be interpreted as the space of Potential games, near Solenoidal games, and non-strategic games.

$$\mathcal{P} = \left\{ u \in C_0^M \mid Du \neq 0, Du \in \ker(d_2) \right\} \cup \{0\},$$
$$\tilde{\mathcal{S}} = \left\{ u \in C_0^M \mid Du \neq 0, Du \in \ker(d_2)^\perp \right\} \cup \{0\}, \quad \mathcal{N} = \left\{ u \in C_0^M \mid Du = 0 \right\}.$$

It is easy to see that the intersection between any two classes is 0.

We first discuss the games that differ by only a non-strategic subgame. Then, we discuss the games with no near Solenoidal parts or no Potential parts in detail.

As implied by its name, the non-strategic games are a class of subgames that do not affect the set of equilibria. Formally, we define the strategic equivalence of Differentiable games.

**Definition 2.6.** *Let $\mathcal{G}$ and $\mathcal{G}'$ be two different Differentiable games that only differ in the utility functions. Let $\{u\}_{i=1}^M$ be the utility functions of $\mathcal{G}$ and $\{u'\}_{i=1}^M$ be the utility functions of $\mathcal{G}'$. $\mathcal{G}$ and $\mathcal{G}'$ are said to be strategically equivalent, if for any $i \in [M]$, $u_i(\omega_i', \omega_{-i}) - u_i(\omega_i, \omega_{-i}) = u_i'(\omega_i', \omega_{-i}) - u_i'(\omega_i, \omega_{-i})$, for any strategy $\omega_i, \omega_i', \omega_{-i}$.*

Strategically equivalent games have the same payoff comparisons per player, and hence have the same set of Nash equilibria. The non-strategic set of $\mathcal{N}$ helps characterize strategically equivalent games through the following lemma.

**Lemma 2.1.** *Two Differentiable games are strategically equivalent if their difference is a non-strategic game.*

*Proof.* The proof is identical to that of Candogan et al. (2011); Legacci et al. (2024) and hence we omit it here. □

## 2.4 THE POTENTIAL GAME

One of the most popular solution concepts in multi-agent settings is Nash equilibrium, which originates from game theory to describe the behaviors of rational, selfish players Roughgarden (2010). It characterizes a stable state between the players, where no individual has any incentive to unilaterally deviate from their chosen strategy. It is worth noting that the computational complexity of finding a Nash equilibrium in a general game is known to be PPAD-Hard Chen et al. (2009); Daskalakis (2013). Nonetheless, it has been established that the computation of a Nash equilibrium becomes more feasible in specific game contexts, such as potential games Monderer & Shapley (1996), where a potential function is available to quantify how individual strategy changes affect collective utility. A long line of works has developed efficient algorithms that can converge to Nash equilibrium in potential games Cominetti et al. (2010); Chen & Lu (2016); Heliou et al. (2017); Cui et al. (2022);

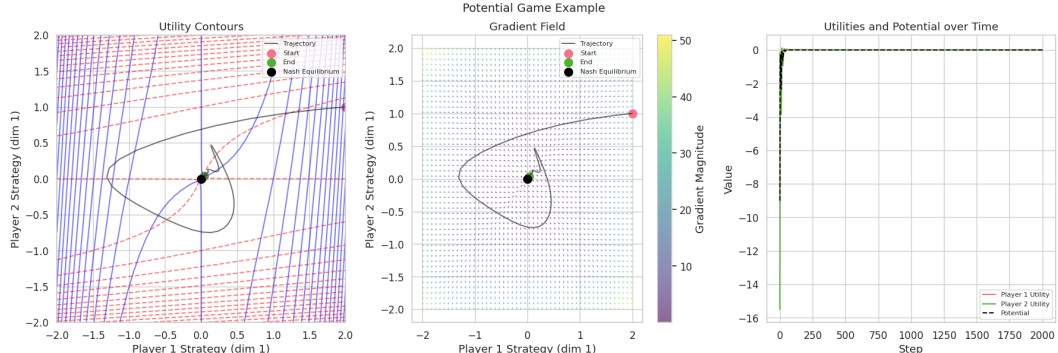

Figure 1: In this two-player potential game, the utility functions are given by $u_1(x, y) = -\frac{1}{2}\|x\|^2 + x^T y - \|x\|^4$ for player 1 and $u_2(x, y) = -\frac{1}{2}\|y\|^2 + x^T y - \|y\|^4$ for player 2, where $x, y \in \mathbb{R}^2$. These utilities are derived from the potential function $P(x, y) = -\frac{1}{4}(\|x\|^2 + \|y\|^2) + \frac{1}{2}x^T y - \frac{1}{4}(\|x\|^4 + \|y\|^4)$. The unique Nash equilibrium is at $(0, 0)$.

Panageas et al. (2023); Anagnostides et al. (2022). The formal definition of a potential game is given as follow.

**Definition 2.7** (Monderer & Shapley (1996)). *A game is a potential game if there is a single potential function $\phi : \mathbb{R}^n \to \mathbb{R}$ and positive numbers $\{\alpha_i > 0\}_{i=1}^M$ such that $\phi(\omega_i', \omega_{-i}) - \phi(\omega_i'', \omega_{-i}) = \alpha_i(u_i(\omega_i', \omega_{-i}) - u_i(\omega_i'', \omega_{-i}))$, for all $i$ and all $\omega_i', \omega_i'', \omega_{-i}$.*

We now show that if a game is a Potential game (without a Solenoidal part), then it has a utility function classified in $\mathcal{P} \oplus \mathcal{N}$ and is an exact potential game.

**Lemma 2.2.** *If $u \in \mathcal{P} \oplus \mathcal{N}$, then the Jacobian matrix $J(\omega) =$*
$$
\begin{pmatrix}
\nabla^2_{\omega_1} u_1 & \nabla^2_{\omega_1,\omega_2} u_1 & \cdots & \nabla^2_{\omega_1,\omega_n} u_1 \\
\nabla^2_{\omega_2,\omega_1} u_2 & \nabla^2_{\omega_2} u_2 & \cdots & \nabla^2_{\omega_2,\omega_n} u_2 \\
\vdots & & & \vdots \\
\nabla^2_{\omega_M,\omega_1} u_M & \nabla^2_{\omega_M,\omega_2} u_M & \cdots & \nabla^2_{\omega_n} u_M
\end{pmatrix}
$$
*is symmetric, and $u$ is an exact potential game.*

*Proof.* We first show that $d_1\phi = Du$ (Lemma B.3), then it is easy to see that the Jacobian matrix is symmetric. Then by Lemma 2 and Corollary 3 of Letcher et al. (2019), we have the equivalence to an exact potential game. $\square$

While it is long known that the gradient descent dynamic is convergent in Potential games, we provide the Figure 1 of the gradient descent dynamic to contrast with the dynamic in Solenoidal games (which is discussed in the next section).

### 2.5 THE NEAR SOLENOIDAL GAME

Now we turn our attention to the near Solenoidal game, which is a game with a divergence-free gradient field. We took the name Solenoidal from physics, where it refers to a field that is divergence-free. The divergence of $X = (g_1, \ldots, g_n)$ is defined as $\mathrm{div}(X) = \sum_{i=1}^n \frac{\partial g}{\partial x_i}$.

**Lemma 2.3.** *For $u \in \tilde{\mathcal{S}} \oplus \mathcal{N}$, and each $u_i \in C_c^\infty(\mathbb{R}^n), \forall i \in [M]$, then $\mathrm{div}(Du) = 0$.*

*Proof.* The result follows from the definition of $\tilde{\mathcal{S}}, \mathcal{N}$, and Lemma D.1. $\square$

We refer to the class of games with divergence-free individual gradient vector fields as Solenoidal games. However, we formally term the games in $\tilde{\mathcal{S}} \oplus \mathcal{N}$ as near Solenoidal games, as we have Lemma 2.3 under the assumption that each $u_i \in C_c^\infty(\mathbb{R}^n)$. In the general case, when each $u_i$

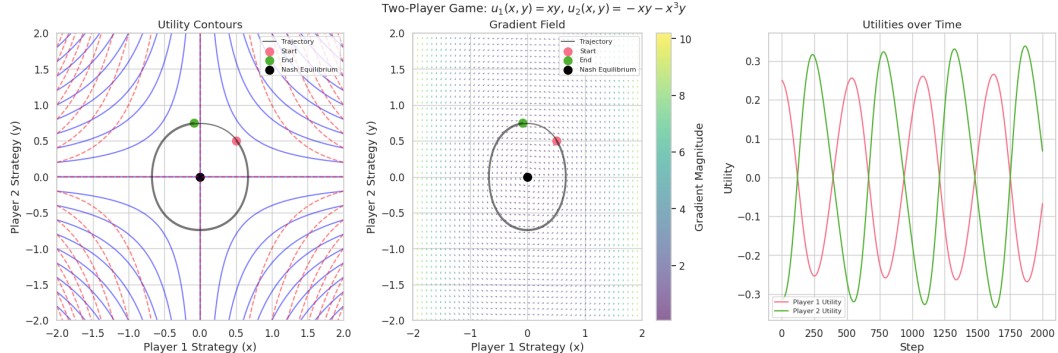

Figure 2: The dynamics of gradient descent, changes of utility over time and gradient vector field of the game presented in Example 2.1.

does not have compact support, we only have the guarantee of $\mathrm{div}(Du) + \sum_{j=1}^{n} \frac{\partial^2 \mathrm{div}(Du)}{\partial^2 x_j} = 0$. Note that when $u$ is compactly supported, i.e. each $u_i$ is compactly supported, then we can have $\mathrm{div}(Du) = 0$, which represents an exact Solenoidal game. In Section 2.7, we introduce a different decomposition scheme that gives an exact Solenoidal game.

While the learning dynamics are well understood in Potential games, their behaviors remain unclear in Solenoidal games. We consider the dynamical system defined by ascending according to the direction of the gradients, i.e.

$$\dot{x} = Du. \tag{1}$$

We first provide an example where gradient descent dynamics is orbiting in a game with zero divergence of the gradient vector field.

**Example 2.1.** *Consider a two-player game where player 1 plays strategy $x \in \mathbb{R}$ and player 2 plays strategy $y \in \mathbb{R}$. The utility functions are given as $u_1(x, y) = xy$, $u_2(x, y) = -xy - x^3y$, One can check that the gradient is $Du = (y, -x - x^3)$ and $\mathrm{div}(Du) = 0$. Moreover, the orbits of the gradient descent are solutions to the curves $\frac{y^2}{2} + \frac{x^2}{2} + \frac{x^4}{4} = c^2$, where $c$ is a constant.*

Figure 2 visually illustrates the orbiting behaviors of gradient descent in a two-player game described Example 2.1.

To investigate the behavior of this dynamic system described above. We start with some definitions of flows on $\mathbb{R}^n$. We refer to Lee & Lee (2012) for a general theory and Legacci et al. (2024) (Appendix A) for a more comprehensive introduction.

Consider a smooth vector field $X$ on $\mathbb{R}^n$, a smooth global integral curve of $X$ is a smooth curve $x : \mathbb{R} \to \mathbb{R}^n$ such that $\dot{x}(t) = X(x(t)), \forall t \in \mathbb{R}$. If a smooth global integral $x$ with starting point $y$ exists, then it is the unique maximal solution of $x(0) = y$, $\dot{x}(t) = X(x(t))$. A smooth global flow on $\mathbb{R}^n$ is a smooth map $\theta : \mathbb{R} \times \mathbb{R}^n \to \mathbb{R}^n$ such that $\forall t, s \in \mathbb{R}$ and $x \in \mathbb{R}^n$, $\theta(0, x) = x$ and $\theta(t, \theta(s, x)) = \theta(t + s, x)$. Fix some $t \in \mathbb{R}$, the orbit map $\theta_t : \mathbb{R}^n \to \mathbb{R}^n$ is $\theta_t(x) = \theta(t, x)$. Fix $x \in \mathbb{R}$, the curve $\theta^x : \mathbb{R} \to \mathbb{R}^n$ denotes $\theta^x(t) = \theta(t, x)$.

For any open set $\mathcal{U} \subseteq \mathbb{R}^n$ and $t \in \mathbb{R}$, $\mathcal{U}_t$ is defined to be image of $\mathcal{U}$ under the orbit map $\theta_t$, i.e. $\mathcal{U}_t = \theta_t(\mathcal{U}) = \{\theta_t(x) : x \in \mathcal{U}\} \subseteq \mathbb{R}^n$.

**Theorem 2.4** (Euclidean Liouville's theorem). *Given a smooth vector field $X$ in $\mathbb{R}^n$ and an open set $\mathcal{U} \subseteq \mathbb{R}^n$, $\frac{d}{dt} \mathbf{vol}(\mathcal{U}_t) = \int_{\mathcal{U}_t} \mathrm{div}(X)dx$, $\forall t \in \mathbb{R}$ such that the flow of $X$ is defined.*

We say a map $\phi : \mathbb{R}^n \to \mathbb{R}^n$ is volume preserving if $\mathbf{vol}(\mathcal{U}) = \mathbf{vol}(\phi\mathcal{U})$, $\forall \mathcal{U} \subseteq \mathbb{R}^n$. An important implication of Liouville's theorem is that orbit maps of vector fields with zero divergence are volume-preserving.

**Corollary 2.1.** *If a vector field $X$ in $\mathbb{R}^n$ has $\mathrm{div}(X) = 0$, then $\mathbf{vol}(\mathcal{U}_t) = \mathbf{vol}(\mathcal{U})$, $\forall \mathcal{U} \subset \mathbb{R}^n$, $\forall t \in \mathbb{R}$ such that the flow of $X$ is defined*

To discuss the behaviors of the dynamical system defined in $X$, we maintain the following assumption of the dynamical system. We first remark that this assumption is required from an ergodic theory perspective Bekka & Mayer (2000), and then provide an example to illustrate the necessity of it.

**Lemma 2.4.** *If $Du$ is compactly supported on $\Omega$, then Assumption C.1 is satisfied.*

Given a measure space $(\Omega, \mu)$, we say that $(\Omega, \mu)$ is finite if $\mu(\Omega) < \infty$, and that a map $\phi : \Omega \to \Omega$ is measure preserving if $\mu(\phi\mathcal{U}) = \mu(\mathcal{U})$ for all measurable subsets $\mathcal{U} \subseteq \Omega$.

**Theorem 2.5** (Poincaré - Measure setting Bekka & Mayer (2000))**.** *Let $(\Omega, \mu)$ be a finite measure space, and let $\phi : \Omega \to \Omega$ be a measure preserving mapping. Let $\mathcal{U}$ be a measurable subset of $\Omega$. Then almost every point $x \in \mathcal{U}$ is infinitely recurrent with respect to $\mathcal{U}$, that is, the set $\{n \in \mathbb{N} : \phi^n x \in \mathcal{U}\}$ is infinite.*

Under our assumption, we obtain the following Lemma.

**Lemma 2.5.** *Let $(\Omega, \mu)$ be a Lebesgue measure space. Then, under Assumption C.1, $\theta_t$ is measure-preserving.*

**Theorem 2.6.** *If a game is near Solenoidal, i.e. $Du \in \tilde{\mathcal{S}} \oplus \mathcal{N}$, and $u$ is compactly supported on $\Omega$, i.e. each $u_i$ is compactly supported, $i \in [M]$, then the system defined in Equation (1) is a poincaré recurrent. Specifically, for almost every initialization $x(0) \in \Omega$, the induced trajectory $x(t)$ returns arbitrarily close to $x(0)$ infinitely often.*

*Proof.* If $u$ has compact support and $u \in \tilde{\mathcal{S}} \oplus \mathcal{N}$, so does $Du$. Then $Du$ satisfies Assumption C.1. By Lemma 2.3 and Theorem 2.4, 2.5, the statement follows. $\square$

### 2.6 Connection to harmonic and Hamiltonian games

To illustrate the relationship between the Solenoidal games and the Hamiltonian games Letcher et al. (2019), we first provide an example of a Solenoidal game that is not Hamiltonian.

**Example 2.2** (Solenoidal games may not be Hamiltonian)**.** *Consider a two-player game where player $1$ plays strategy $x \in \mathbb{R}$ and player $2$ plays strategy $y \in \mathbb{R}$. The utility functions are given as $u_1(x,y) = x^2$, $u_2(x,y) = -y^2$. In this case, the gradient field is $Du = (2x, -2y)$ and therefore $\text{div}(Du) = 0$, so the game is a Solenoidal game. However, the Jacobian matrix $\begin{pmatrix} 2 & 0 \\ 0 & -2 \end{pmatrix}$ is not skew-symmetric and is therefore not Hamiltonian.*

However, we provide the following lemma to show that Hamiltonian games are Solenoidal games.

**Lemma 2.6.** *A Hamiltonian game is a Solenoidal game.*

*Proof.* If a game with utility $u$ is Hamiltonian, then the diagonal of the Jacobian matrix is $0$. This implies $\text{div}(Du) = 0$ by the definition of divergence. $\square$

For a finite game, through Helmholtz decomposition, it can be decomposed into a potential part, which is the same as our Potential part, and a harmonic part. In Differentiable games, however, the harmonic part does not exist, as the harmonic part is $\ker(\Delta_1) = \ker(d_2) \cap \ker(d_1^*) = 0$, as $\text{im}(d_1) = \ker(d_2)$. This is because of the fact that the harmonic component is isomorphic to the de Rham cohomology of the manifold, which is zero when the differential $k$-form is with $k = 1$ and the manifold is $\mathbb{R}^n$. Despite the absence of harmonic games, the Solenoidal game of the Differentiable games exhibits properties similar to those of harmonic games, such as the divergence-free gradient vector field. In fact, this is the key property that leads to the chaotic behavior of learning algorithms in harmonic games.

### 2.7 Alternative decomposition

In the remarks following Lemma 2.3, we mentioned that the decomposition presented so far only yields a direct sum decomposition of an exact Potential game and a near Solenoidal game. To bridge the gap between near and exact Solenoidal games, we operated under the assumption that the utility function is compactly supported to show the non-convergent behaviors in the Solenoidal

game. In this section, we discuss an alternative decomposition, which gives a decomposition of a near Potential game and an exact Solenoidal game. To obtain this decomposition, we would define a slightly different $C_1$. Due to the space limit, We defer the specific definition to the appendix.

As we decompose to an exact Solenoidal game, we can show the non-convergent behaviors in the Solenoidal game without additional assumptions. Beyond the recurrence, the Nash equilibrium point is also nonconvex and nonconcave in exact Solenoidal games. Hence, this property shows new insights into the challenges of finding Nash equilibrium in Differentiable games with second-order optimization methods. One can choose either one, out of our two decompositions, according to their specific game.

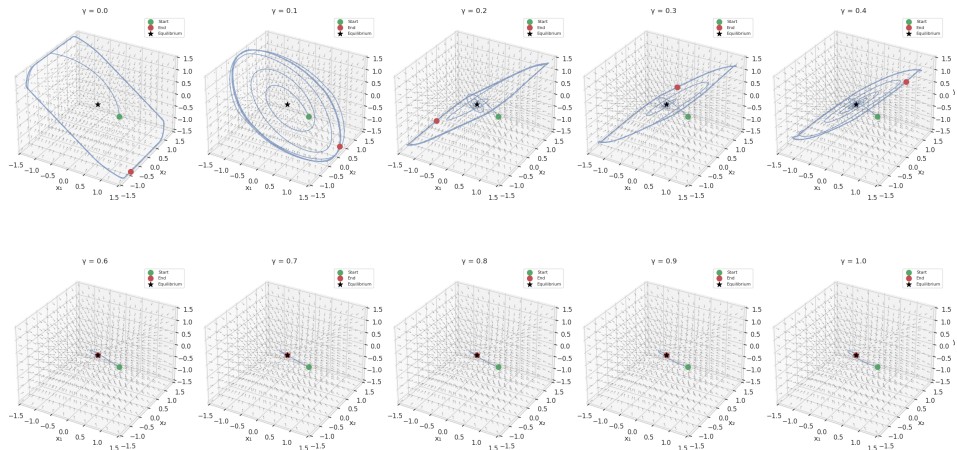

Figure 3: The strategies $x, y \in \mathbb{R}^2$, $x = (x_1, x_2), y = (y_1, y_2)$ in a spectrum of games by interpolating scalar and Solenoidal games. The utility functions for the Potential game are $U_1 = -((x_1 - y_1)^2 + (x_2 - y_2)^2) - 0.2(x_1^2 + x_2^2)$, $U_2 = -((x_1 - y_1)^2 + (x_2 - y_2)^2) - 0.2(y_1^2 + y_2^2)$, while the utility for the Solenoidal games are $U_1 = x_1 y_2 - x_2 y_1 U_2 = -(x_1 y_2 - x_2 y_1) - 0.1(x_1^2 y_2^2 + x_2^2 y_1^2)$.

## 3  INTERPOLATION OF SCALAR AND SOLENOIDAL GAMES

Our decomposition method can interpolate the Differentiable games on a spectrum, where the exact scalar and Solenoidal games are at the two ends. In most of the applications, the Differentiable game considered is a mixture of the two games. While it is hard to obtain theoretical guarantees of the learning dynamics for the games in the middle of the spectrum, we provide the following figure to show the behaviors of gradient descent on the interpolations of scalar and Solenoidal games through simulation. The utility of the interpolated game is a convex combination of the scalar and Solenoidal game, $u = \gamma u_{SP} + (1 - \gamma) u_{VP}$, where $u_{SP}$ is the utility function of a Potential game and $u_{VP}$ is the utility function of a Solenoidal game. The parameter $\gamma$ varies from 0 to 1, allowing for a smooth transition between the two types of games.

## 4  CONCLUSION AND FUTURE WORKS

In this work, we have provided two decompositions of Differentiable games through the Hodge/Helmholtz decomposition. In summary, we decompose a Differentiable game into a Potential game and a Solenoidal game. We showed that the Potential game is an exact potential game introduced by Monderer & Shapley (1996), where the gradient descent dynamic can effectively find the Nash equilibrium. For the Solenoidal game, we showed that the individual gradient field is divergence-free, which means that the gradient descent dynamic may either diverge or exhibit recurrent behavior. Technically, we introduced the Sobolev space to ensure that the gradient vector field is a Hilbert space, thereby achieving a direct sum decomposition of the simultaneous gradient field levels in Differentiable games.

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

# A    PROOF FOR THEOREM 2.3

We first show that the gradient vector fields are in $C_1$.

**Lemma A.1.** *For any $m \in [M]$, we have $Du \in C_1$, i.e. $(Du)_m \in H^1(\mathbb{R}^n)$.*

*Proof.* As $u_m \in C_0$, by the definition of the inner product, we have

$$\|u_m\|_0^2 = \int_{\mathbb{R}^n} |u_m|^2 dx + \sum_{i=1}^n \int_{\mathbb{R}^n} \left|\frac{\partial u_m}{\partial x_i}\right|^2 dx$$

$$+ \sum_{i=1}^n \sum_{j=1}^n \int_{\mathbb{R}^n} \left|\frac{\partial^2 u_m}{\partial x_i \partial x_j}\right|^2 dx$$

$$< \infty.$$

Therefore, $\left\|\frac{\partial u_m}{\partial x_i}\right\|_{L^2}^2 < \infty, \left\|\frac{\partial^2 u_m}{\partial x_i \partial x_j}\right\|_{L^2}^2 < \infty$. Hence $Du \in C_1$.    □

To obtain a decomposition of $C_1$, it remains to show that $d_2$ is a bounded linear operator, which is guaranteed by leveraging the inner products defined on $C_0$, $C_1$.

**Lemma A.2.** *$d_2$ is a bounded linear operator.*

*Proof.* By the definition of $d_2$, it is clearly linear. To show that it is bounded, we need to show that there exists some constant $c$ such that $\|d_2 X\|_2 \leq c\|X\|_1$. Using the inequality of $(a-b)^2 \leq 2a^2 + 2b^2$, we have

$$\|d_2 X\|_2^2 = \sum_{i<j} \int_{\mathbb{R}^n} \left|-\frac{\partial f_i}{\partial x_j} + \frac{\partial f_j}{\partial x_i}\right|^2 dx$$

$$\leq 2\left(\sum_{i<j} \int_{\mathbb{R}^n} \left|\frac{\partial f_i}{\partial x_j}\right|^2 + \left|\frac{\partial f_j}{\partial x_i}\right|^2 dx\right)$$

$$\leq 4\|X\|_1^2,$$

where the last inequality is by noticing the second term of the definition of $\|\cdot\|_1$. Taking the square root of both sides shows that $d_2$ is bounded.    □

**Theorem 2.3.** *$C_1$ can be decomposed as $C_1 = \ker(d_2) \oplus \ker(d_2)^\perp$. Then $Du = X_{\mathcal{P}} + X_{\tilde{\mathcal{S}}}$, where $X_{\mathcal{P}} \in \ker(d_2)$, $X_{\tilde{\mathcal{S}}} \in \ker(d_2)^\perp$.*

*Proof.* As $d_2$ is bounded, we have $\overline{\ker(d_2)} = \ker(d_2)$. Then, we obtain the result by noticing $C_1 = \overline{\ker(d_2)} \oplus \ker(d_2)^\perp$. By Lemma A.1, we can decompose $Du$ as described.    □

# B    PROOF FOR LEMMA 2.2

**Lemma B.1.** *If $X \in \ker(d_2)$, we say it is closed; if $X \in \text{im}(d_1)$, we say it is exact. Then, every exact $X$ is closed, i.e. $d_2 d_1 = 0$.*

*Proof.* By the definition of $d_1$, we have $d_1 f = \left(\frac{\partial f}{\partial x_1}, \dots, \frac{\partial f}{\partial x_n}\right)$. Then applying $d_2$ yields

$$(d_2 d_1 f)_{ij} = -\frac{\partial^2 f}{\partial x_j x_i} + \frac{\partial^2 f}{\partial x_i x_j} = 0.$$

□

**Lemma B.2** (Poincaré Lemma). *In $\mathbb{R}^n$, every smooth closed 1-form is exact with $n > 1$.*

**Lemma B.3.** *If $u \in \mathcal{P} \oplus \mathcal{N}$, then there exist some $\phi \in C^\infty(\mathbb{R}^n)$, such that $d_1 \phi = Du$.*

*Proof.* By the definition of $\mathcal{P}$, $\mathcal{N}$, we have $Du \in \ker(d_2)$ and is therefore closed. Since $u \in C_0^M$, $Du$ is also smooth. Thus by the application of Lemma B.2 and Lemma B.1, we have the result.    □

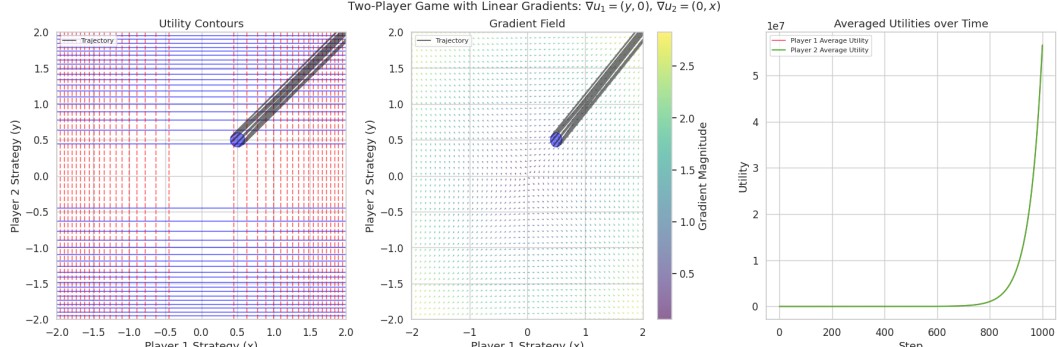

Figure 4: The dynamics of gradient descent, changes of utility over time, and gradient vector field of a two-player game with utility gradients $Du = (y, x)$. We plotted the trajectory that started from a set of initial points. We can see that the volume of the set remains unchanged throughout the trajectory.

## C   BOUNDED ORBIT ASSUMPTION

**Assumption C.1** (Bounded orbit). *There exists a compact set $\Omega$ such that for any $x_0 \in \Omega$, $\theta_t(x_0) \in \Omega$, $\forall t \geq 0$.*

We now explain the necessity and implications of this assumption. Consider the case that there exists some $t_1$ such that $\theta_{t_1}(x_0) \notin \Omega$, then we consider a set $\Omega_1$, $\Omega \subseteq \Omega_1$, $2\,\mathbf{vol}(\Omega) \leq \mathbf{vol}(\Omega_1)$, $\theta_{t_1}(x_0) \in \Omega_1$. If there exists some $t_2 > t_1$, $\theta_{t_2}(x_0) \notin \Omega_1$. Then we consider some set $\Omega_2$, $\Omega_1 \subseteq \Omega_2$, $2\,\mathbf{vol}(\Omega_1) \leq \mathbf{vol}(\Omega_2)$, $\theta_{t_2}(x_0) \in \Omega_2$. By repeatedly applying this argument, we can see that the trajectory $x(t)$ induced by $x_0$ can go to infinity, and the dynamical system can be considered divergent.

Without this assumption, a simple game where the trajectory of gradient descent goes to infinity is a two-player game with $Du = (y, x)$. We visually illustrate the dynamic in this game in Figure 4.

## D   PROOF FOR THE NEAR SOLENOIDAL GAME

**Lemma D.1.** *If $X \in (\ker(d_2))^{\perp}$, $X = (g_1, \ldots, g_n)$ and each $g_i \in H_c^3(\mathbb{R}^n) := \overline{C_c^{\infty}(\mathbb{R}^n)}^{\|\cdot\|_{H^3}}$, then $\mathrm{div}(X) = 0$ almost everywhere.*

*Proof.* For any $f \in H_c^2(\mathbb{R}^n)$, we have $d_2 d_1 f = 0$, then $d_1 f \in \ker(d_2)$. Hence, by the definition of inner products on $C_1$, we have

$$0 = \langle d_1 f, X \rangle_1$$

$$= \sum_{i=1}^n \int_{\mathbb{R}^n} \frac{\partial f}{\partial x_i} \cdot g_i dx + \sum_{i=1}^n \sum_{j=1}^n \int_{\mathbb{R}^n} \frac{\partial^2 f}{\partial x_j \partial x_i} \cdot \frac{\partial g_i}{\partial x_j} dx$$

$$= -\left( \sum_{i=1}^n \int_{\mathbb{R}^n} f \cdot \frac{\partial g_i}{\partial x_i} dx + \sum_{i=1}^n \sum_{j=1}^n \int_{\mathbb{R}^n} \frac{\partial f}{\partial x_j} \cdot \frac{\partial^2 g_i}{\partial x_j \partial x_i} dx \right)$$

$$= -\left( \int_{\mathbb{R}^n} f \cdot \mathrm{div}(X) + \sum_{j=1}^n \int_{\mathbb{R}^n} \frac{\partial f}{\partial x_j} \cdot \frac{\partial \mathrm{div}(X)}{\partial x_j} dx \right).$$

As $\mathrm{div}(X) \in H_c^2(\mathbb{R}^n)$, substituting it with $f$, we have $\mathrm{div}(X) = 0$ almost everywhere.   □

**Lemma 2.4.** *If $Du$ is compactly supported on $\Omega$, then Assumption C.1 is satisfied.*

*Proof.* Suppose that the claim does not hold, then there exists an $x \in \Omega$, $t_1 > 0$, $x_1 = \theta^x(t_1) = \theta_{t_1}(x) \notin \Omega$, where $\Omega$ is a compact set. Let $\tilde{t} = \sup\{t \mid \theta^x(t) \in \Omega, t \leq t_1\}$, $\tilde{x} = \theta^x(\tilde{t})$. Since $\theta^x(t)$ is continuous, we have $\tilde{x} \in \Omega$. By the mean value theorem, there exists a $t_2 \in (\tilde{t}, t_1)$ such that $Du(\theta^x(t_2)) = \frac{d\theta^x}{dt}(t_2) = \frac{x_1 - \tilde{x}}{t_1 - \tilde{t}} \neq 0$. By the definition of $\tilde{t}$, $\theta^x(t_2) \notin \Omega$. Then we have $Du(\theta^x(t_2)) = 0$, which indicates that the claim holds by proof by contradiction. $\square$

**Lemma 2.5.** *Let $(\Omega, \mu)$ be a Lebesgue measure space. Then, under Assumption C.1, $\theta_t$ is measure-preserving.*

*Proof.* By Assumption C.1, $\forall \mathcal{U} \in \Omega, \tilde{\mathcal{U}}_t = \mathcal{U}_t \cap \Omega = \mathcal{U}_t$. Then, by Theorem 2.4 $\mu(\tilde{\mathcal{U}}_t) = \mu(\mathcal{U})$. $\square$

# E ALTERNATIVE DECOMPOSITION

**Definition E.1.** *The Hilbert space $\tilde{C}_1$ is defined as $\tilde{C}_1 = \left\{ X = (f_1, \ldots, f_n) \mid f_i \in L^2(\mathbb{R}^n) \right\}$. The inner product is defined as $\langle X, Y \rangle_1 = \sum_{i=1}^{n} \int_{\mathbb{R}^n} f_i \cdot g_i dx$. The norm is thus $\|X\|_1 = [\langle X, X \rangle_1]^{1/2}$.*

We then define the operator that maps between $C_0$ and $\tilde{C}_1$ as $\tilde{d}_1$.

**Definition E.2.** *The operator $\tilde{d}_1 : C_0 \to \tilde{C}_1$ is defined as follows. For $f \in C_0$, $\tilde{d}_1 f = \left( \frac{\partial f}{\partial x_1}, \ldots, \frac{\partial f}{\partial x_n} \right)$.*

In this decomposition, we drop the condition that the utility has to be compactly supported. Instead, we assume that each $u_m$ is in the following class of $\tilde{C}_0^M$, where $\tilde{C}_0^M = \{u = (u_1, \ldots, u_M) \mid u_i \in C_0, \forall i\}$.

Equipped with these results, we show the second decomposition.

**Theorem E.1.** *$\tilde{C}_1$ can be decomposed as $\tilde{C}_1 = \overline{\text{im}(\tilde{d}_1)} \oplus \text{im}(\tilde{d}_1)^{\perp}$. Then $Du = X_{\tilde{\mathcal{P}}} + X_{\mathcal{S}}$, where $X_{\tilde{\mathcal{P}}} \in \overline{\text{im}(\tilde{d}_1)}$, and $X_{\mathcal{S}} \in \text{im}(\tilde{d}_1)^{\perp}$.*

*Proof.* The result follows immediately from the decomposition of $\tilde{C}_1$ and Lemma F.1. $\square$

Now we can define the following classes of subgames, which can be interpreted as the space of the near Potential games, the exact Solenoidal games, and non-strategic games.

$$\tilde{\mathcal{P}} = \left\{ u \in \tilde{C}_0^M \mid Du \neq 0, Du \in \overline{\text{im}(\tilde{d}_1)} \right\} \cup \{0\},$$

$$\mathcal{S} = \left\{ u \in \tilde{C}_0^M \mid Du \neq 0, Du \in \text{im}(\tilde{d}_1)^{\perp} \right\} \cup \{0\},$$

$$\tilde{\mathcal{N}} = \left\{ u \in C_0^M \mid u \in \ker(D) \right\}.$$

We say the games in $\tilde{\mathcal{P}} \oplus \tilde{\mathcal{N}}$ are near potential games, as we can only find an $\epsilon$-potential function. Specifically, we have the following corollary for the near potential games.

**Corollary E.1.** *If $u \in \tilde{\mathcal{P}} \oplus \tilde{\mathcal{N}}$, then for any $\epsilon > 0$, there exist some $\phi \in C_0$, such that $\|\tilde{d}_1 \phi - Du\|_1 \leq \epsilon$.*

The learning dynamics of finite near potential games have been extensively investigated in Candogan et al. (2010; 2013a;b). While it is unclear whether the results on the finite near potential game can be immediately extended to differential near potential games, it is reasonable to conjecture that they might behave similarly to that of an exact potential game. We leave it as future work to investigate the dynamics of differential near potential games.

For the exact Solenoidal game, we show that the divergence of the gradient field is exactly zero.

As one might expect, the dynamical system (1) by gradients ascending exhibits similar behavior in Solenoidal games as is in near Solenoidal games. Namely, by the divergence-free result, it is either poincaré recurrent or divergent. Notice that, though, in Solenoidal games it no longer requires a compactly supported utility function.

**Theorem E.2.** *If a game is a Solenoidal game, i.e. $Du \in \mathcal{S} \oplus \tilde{\mathcal{N}}$, and Assumption C.1 is satisfied, Equation (1) is a poincaré recurrent. Specifically, for almost every initialization $x(0) \in \Omega$, the induced trajectory $x(t)$ returns arbitrarily close to $x(0)$ infinitely often.*

*Proof.* By Lemma F.2 and Theorem 2.4, 2.5, we have the result. $\square$

Beyond the recurrence, the Nash equilibrium point is also nonconvex and nonconcave in exact Solenoidal games. Hence, this property shows new insights into the challenges of finding Nash equilibrium in Differentiable games with second-order optimization methods.

**Lemma E.1.** *If $\operatorname{div}(Du) = 0$, and $p$ is a local Nash equilibrium in $\mathcal{U}$, i.e. $u_m(p) \geq u_m(q^m, p^{-m})$, $\forall m, q^m \in \mathcal{U}$, then $\frac{\partial^2 u_m}{\partial^2 \omega_i^m}(p) = 0$.*

*Proof.* By the definition, we have $\operatorname{div}(Du) = \sum_{m=1}^{M} \sum_{i=1}^{n} \frac{\partial^2 u_m}{\partial \omega_i^m} = 0$, and each $\frac{\partial^2 u_m}{\partial^2 \omega_i^m}(p) \leq 0$ as $p$ is a local Nash equilibrium. Therefore each $\frac{\partial^2 u_m}{\partial^2 \omega_i^m}$ must be 0. $\square$

# F    Proof for Alternative Decomposition

**Lemma F.1.** *For any $m \in [M]$, $Du \in \tilde{C}_1$ and $(Du)_m \in L^2(\mathbb{R}^n)$.*

*Proof.* The proof is similar to that of A.1, which we omit here. $\square$

**Lemma F.2.** *If $X \in \operatorname{im}(\tilde{d}_1)^{\perp}$, $X = (g_1, \ldots, g_n)$, and $g_i \in H^1(\mathbb{R}^n)$ for each $i$, then $\operatorname{div}(X) = 0$ almost everywhere.*

*Proof.* For any $f \in C_c^{\infty}(\mathbb{R}^n)$, as $X \in \operatorname{im}(\tilde{d}_1)^{\perp}$, $\langle \tilde{d}_1 f, X \rangle_1 = 0$. Then

$$0 = \sum_{i=1}^{n} \int_{\mathbb{R}^n} \frac{\partial f}{\partial x_i} \cdot g_i dx$$

$$= \sum_{i=1}^{n} \left( - \int_{\mathbb{R}^n} f \cdot \frac{\partial g_i}{\partial x_i} \right)$$

$$= - \int_{\mathbb{R}^n} f \cdot \operatorname{div}(X) dx.$$

By Lemma F.3, we have $\operatorname{div}(X) = 0$ almost everywhere. $\square$

**Lemma F.3.** *For any $f \in C_C^{\infty}(R^n)$, $g \in L^2(\mathbb{R}^n)$, if $\int_{R^n} f(x)g(x)dx = 0$, then $g(x) = 0$ almost everywhere.*

*Proof.* By Corollary 4.2 of Book 2 in Brezis (2011), $C_c^{\infty}(R^n)$ is dense in $L^2(R^n)$. There exists $g_k \in C_c^{\infty}(R^n)$ such that $\|g_k - g\|_{L^2(R^n)} \longrightarrow 0$. Therefore $\int_{R^n} g_k(x)g(x)dx = 0$, and

$$\left| \int_{R^n} g(x) \cdot g(x)dx - \int_{R^n} g_k(x) \cdot g(x)dx \right|$$

$$\leq \int_{R^n} |g_k - g| \cdot |g| dx$$

$$\leq \|g_k - g\|_{L^2(R^n)} \cdot \|g\|_{L^2(R^n)} \xrightarrow{\text{as } k \to \infty} 0.$$

As

$$\int_{R^n} |g(x)|^2 dx = \lim_{k \to \infty} \int_{R^n} g_k(x)g(x)dx = 0,$$

we have $g(x) = 0$ almost everywhere. $\square$

