# OpenReview forum: "On the Decomposition of Differentiable Games"
_ICLR.cc/2026/Conference — Submitted to ICLR 2026_

### Official Review · Reviewer_vot3 · 2025-10-26

**Soundness:** 3
**Presentation:** 3
**Contribution:** 2
**Rating:** 2
**Confidence:** 5

**Summary:**

The authors consider a class of games with continuous (and differentiable payoffs) and consider a corresponding Sobolev space of games.
Using Hodge theorem and related orthogonal decomposition the authors propose to decompose games into direct sum of potential game (i.e gardient flows)  and their complement as well as decomposing divergence free vector fields and theor orthogonal complement.
Some conclusions about the dynamics (namely Poincaré recurrence) are are also proved.

**Strengths:**

a)  The paper is well written and the technical details are explained in a satisfactory manner.

b) The decomposition of games in Sobolev space is novel and actually seems very natural and promising.

c) The problem of understanding and classifying the family of games is a worthy endeavor as game theoretical concepts are now used in a great variety of fields, computer science, economics, evolutionary biology, physics etc...

**Weaknesses:**

a) The conclusion about the game dynamics for divergence free game is very generic (namely Poincare recurrence which is a soft result)
and it is not clear why this is relevant.  Dynamics with divergent free vector fields exhibit an enormous variety of different from fully integrable to strongly mixing.

b) The paper suffers from a lack of concrete examples and/or concrete applications either in economics or computer sciences.  There are for example many games in the economics literature which could be decomposed using the techniques in the papers and it may be quite interesting to analyze these games using these tools.  Alternatively how are the concepts presented here useful in CS applications?

c) There is lack of connections of the results in the papers with the concepts of game theory. I think it is necessary to relate the concept of solenoidal games with more standard game theoretic concepts. For example, how are they related to zero-sum games?

d) I understand the appeal of gradient dynamics but these are not the only possible dynamics in game theory: it would be interesting to understand how the decomposition presented here expresses itself in different contexts.

e) The references to the literature are very incomplete. I am aware of two recent research papers on the decomposition of games with continuous strategy (and there well maybe more).

Sung-Ha Hwang and Luc Rey-Bellet. Strategic decompositions of normal form games: potential games and zero-sum games.
Games and Economic Behavior 122, (2020), pp. 370–390.

and the closely related

Sung-Ha Hwang and Luc Rey-Bellet.
Simple characterizations of potential games and zero-sum equivalent games. Journal of Economic Theory and Econometrics, 31 (2020), pp. 1–13.

These papers have very detailed decomposition of games starting from potential games and zero-sum games (very closely connected to solenoidal game).  The games are decomposed using a different functional analytical framework and using integral rather than differential characterizations, although the second paper also treats differential characterizations.  The paper have two decompositions which look very similar to the decompositions here and the authors push the analysis further by considering finer decompositions.

A detailed comparison of the two approaches (which are slightly different) is in order.

**Questions:**

1) Can you please clarify how your decomposition connects to standard game theoretic concepts, beyond potential games

2) How do your results compare with the existing results in the literature mentioned above?

3) What are possible applications of your results?


I think that the topic is interesting and I am willing to change my rating if my concerns are addressed.

---

### Official Review · Reviewer_ek7J · 2025-10-27

**Soundness:** 3
**Presentation:** 3
**Contribution:** 3
**Rating:** 8
**Confidence:** 3

**Summary:**

This paper addresses a notable open problem posed by Letcher et al. (2019) concerning the extension of the Helmholtz decomposition from finite games to differentiable games with strategy spaces in $\mathbb{R}^n$. The authors employ Sobolev spaces to construct the necessary Hilbert space framework, allowing them to propose two methods for decomposing any differentiable game into a Potential component and a Solenoidal component.

A primary contribution of the paper is linking these components to distinct dynamic behaviors. The Potential part aligns with potential games where gradient dynamics typically converge. The Solenoidal part, characterized by a divergence-free gradient field, is associated with non-convergent, Poincaré recurrent dynamics.

**Strengths:**

The contributions of this work are noteworthy. It provides a theoretical answer to a previously open question. The mathematical framework appears sound and appropriate for the problem. The dynamical insights offer a formal basis for understanding recurrent or cyclic behaviors often observed in ML applications, providing a new perspective for continuous settings. The inclusion of two distinct decompositions provides flexibility. An additional finding is Lemma E.1, which shows that Nash equilibria in exact Solenoidal games must be flat (zero second-order derivatives), which may inform the analysis of second-order optimization methods.

**Weaknesses:**

While the theoretical contributions are clear, several points could benefit from further clarification or extension. The analysis relies on key assumptions, such as compact support or bounded orbits, to prove recurrence. A discussion on the implications of these assumptions for practical ML models would enhance the paper's applicability. Additionally, the implications of "near" potential and "near" solenoidal components are noted as future work. A high-level discussion of how this "nearness" (e.g., the $\epsilon$ in Corollary E.1) might impact stability or convergence speed could provide a more complete picture of the proposed framework. Finally, the paper is well-motivated by GANs and MARL, though its direct algorithmic implications are not explored in detail. The paper focuses on diagnosing dynamics; a discussion on how this decomposition might inform new algorithm designs, perhaps by extending methods like symplectic gradient adjustment, could be a valuable extension.

**Questions:**

What's the direct algorithmic implications of this decomposition for applications such as GANs, MARL, or federated learning?

---

### Official Review · Reviewer_LYrr · 2025-10-29

**Soundness:** 3
**Presentation:** 3
**Contribution:** 2
**Rating:** 4
**Confidence:** 4

**Summary:**

The paper provides decompositions for Differentiable games using Hodge/Helmholtz decomposition. Basically, a differential gems can be decomposed into Potential part, Solenoidal part, and a non-strategic part. The gradient descent dynamics on the potential part, as defined classically, can lead to NE, but either be divergent or recurrent on the Solenoidal part.

**Strengths:**

1. Concrete proofs on the convergence analysis for gradient decent dynamics on the decomposed games.
2.The paper is well-organized and easy to follow.

**Weaknesses:**

1. Although not trivial, the decomposition results can be regarded as a direct extension from Letcher et al. (2019) and Legacci et al. (2024). Even for the novel part, the near Solenoidal game, its property is not hard to imagine.
2. It would be better if there is some successful application, e.g. new algorithmic design, based on the new decomposition.

**Questions:**

Is there any successful application based on the decomposition results?

---

### Official Review · Reviewer_KVkg · 2025-10-31

**Soundness:** 3
**Presentation:** 1
**Contribution:** 1
**Rating:** 2
**Confidence:** 5

**Summary:**

This paper provides a decomposition theorem for games, in the spirit of the potential-harmonic-nonstrategic decomposition of Candogan et al (2011). The main difference and contribution of the paper is that the authors consider differentiable games, that is, games with continuous action sets and differentiable utility functions, as opposed to the finite game setting of Candogan et al (2011).

The authors' main result can be summarized as follows: any differentiable game with smooth, compactly supported utilities can be written as a sum of a potential, a near-solenoidal, and a non-strategic game. [Here, solenoidal means that the divergence of the game's individual gradient field vanishes, and non-strategic means that the gradient field is zero. In this regard, the solenoidal component mirrors the harmonic component of the decomposition of Candogan et al (2011).]

The authors complement their analysis by comparing their decomposition to the symmetric / skew-symmetric decomposition of the Jacobian of the game due to Letcher et al. (2019), who introduced the notion of a Hamiltonian game (games with skew-symmetric Jacobian). They show in particular that any Hamiltonian game is solenoidal, but the converse need not hold.

**Strengths:**

The setting and general question is an interesting one.

**Weaknesses:**

The reason that I'm giving this paper a low score is that it  is stating several times that a major difficulty and contribution is the fact that the authors are treating games over $\mathbb{R}^n$, which makes it difficult to apply Hodge-theoretic tools and results. To quote the authors: "applying [the Hodge theorem] on $\mathbb{R}^n$ is non-trivial due to the non-compactness of $\mathbb{R}^n$. This is, indeed, a major difficulty, which the authors sidestep by working with games with $C^\infty$-smooth functions and compact support. This means that the payoff functions of the game are equal to zero outside a compact set, and this degradation is smooth.

This is a very limiting assumption: it cannot be used to model standard unconstrained games (like the go-to min-max example $u_1(x_1,x_2) = - u_2(x_1,x_2) = - x_1 x_2$) because this game does not have compact support, and it cannot be used to model constrained games (like normal form games) because, in such games, the payoff functions do not vanish smoothly. This rules out all the toy examples used by the authors for illustration purposes—and, in fact, I am not aware of any application-relevant game satisfying this assumpion.

This is not easy to spot: in L205, the authors define the Hilbert $C_0$ as the closure of $C^\infty(\mathbb{R}^n)$ under the Sobolev L2 norm. This would give the impression that the authors are able to treat unconstrained continuous games, but this is not so. The Sobolev norm is not even defined on all of $C^\infty(\mathbb{R}^n)$ (consider the simple one-player game with $u(x) = x$), so the closure is meaningless. Instead, I strongly suspect that the authors meant to consider there the closure of $C_c^\infty$, that is, games with compactly supported payoff functions: this is further supported by L226 where the authors are considering the intersection of $C_0$ with $C^\infty$ (which would be redundant otherwise), and several other points in the paper, where the authors are talking about compactly supported payoff functions.

The compact support assumption also trivializes Theorem 2.6: if the game's payoff functions are smooth and compactly supported, the trajectories of the gradient dynamics are de facto bounded, so Assumption C.1 in Lemma 2.4 (which, annoyingly, is only stated in the appendix) is itself trivially satisfied. Again, however, I struggle to think of an application-relevant example satisfying the compact support assumption, so this theorem is also of very limited conceptual value.

Finally, there is the issue of writing. The authors rely on the readers having a solid understanding of several advanced differential-geometric concepts (differential forms and Hodge / de Rham theory in particular), which would make the paper inaccessible to a large part of ICLR's audience (even the theory component).

It is for these reasons that I'm giving a "reject" recommendation and "fair" to "poor" scores overall. I do believe the authors' basic decomposition idea via Hodge theory has a lot of merit and is worth pursuing, but the current version of the manuscript is not yet close to clearing the bar of acceptance to a top-tier generalist venue like ICLR. The various low scores are thus intended as a measure of the distance that needs to be covered, and they should not be interpreted as a harsh criticism of the authors' work.

**Questions:**

None.

---

### Meta-Review · Area_Chair_Pob3 · 2026-01-05

**Summary:**

The paper's main result is a decomposition theorem for differentiable games with smooth, compactly supported utilities. It is shown that such games can be written as a sum of a potential, a near-solenoidal (divergence is zero), and a non-strategic part, in the same spirit as in prior works, most notably of Candogan et al. There were a few technical concerns that were raised by two reviewers that challenged the actual technical meat and novelty of the work. The authors failed to respond to these concerns and as a result we recommend a rejection of this work.

**Reviewer Concerns:**

KVkg raised the question about the compact support assumption and that it trivializes the result.

**Reviewer Scores:**

Not applicable as there was no rebuttal.

---

### Decision · Program_Chairs · 2026-01-26

Reject